# Research on Enhanced Detection of Benzoic Acid Additives in Liquid Food Based on Terahertz Metamaterial Devices

**DOI:** 10.3390/s21093238

**Published:** 2021-05-07

**Authors:** Jun Hu, Rui Chen, Zhen Xu, Maopeng Li, Yungui Ma, Yong He, Yande Liu

**Affiliations:** 1School of Mechatronics & Vehicle Engineering, East China Jiaotong University, Nanchang 330013, China; hujun_ecjtu@163.com (J.H.); xz2910845707@163.com (Z.X.); geiliofan@163.com (M.L.); 2Department of Optoelectronic Information Engineering, Zhejiang University, Hangzhou 310027, China; chen_rui@zju.edu.cn (R.C.); yungui@zju.edu.cn (Y.M.); 3School of Mechanical Engineering, Zhejiang University, Hangzhou 310027, China; yongqin@zju.edu.cn

**Keywords:** THz detection technology, metamaterials, trace additives, resonance enhancement, LS-SVM

## Abstract

It is very important for human health to supervise the use of food additives, because excessive use of food additives will cause harm to the human body, especially lead to organ failures and even cancers. Therefore, it is important to realize high-sensibility detection of benzoic acid, a widely used food additive. Based on the theory of electromagnetism, this research attempts to design a terahertz-enhanced metamaterial resonator, using a metamaterial resonator to achieve enhanced detection of benzoic acid additives by using terahertz technology. The absorption peak of the metamaterial resonator is designed to be 1.95 THz, and the effectiveness of the metamaterial resonator is verified. Firstly, the original THz spectra of benzoic acid aqueous solution samples based on metamaterial are collected. Secondly, smoothing, multivariate scattering correction (MSC), and smoothing combined with first derivative (SG + 1 D) methods are used to preprocess the spectra to study the better spectral pretreatment methods. Then, Uninformative Variable Elimination (UVE) and Competitive Adaptive Reweighted Sampling (CARS) are used to explore the optimal terahertz band selection method. Finally, Partial Least Squares (PLS) and Least square support vector machine (LS-SVM) models are established, respectively, to realize the enhanced detection of benzoic acid additives. The LS-SVM model combined with CARS has the best effect, with the correlation coefficient of prediction set (R_p_) is 0.9953, the root mean square error of prediction set (RMSEP) is 7.3 × 10^−6^, and the limit of detection (LOD) is 2.3610 × 10^−5^ g/mL. The research results lay a foundation for THz spectral analysis of benzoic acid additives, so that THz technology-based detection of benzoic acid additives in food can reach requirements stipulated in the national standard. This research is of great significance for promoting the detection and analysis of trace additives in food, whose results can also serve as a reference to the detection of antibiotic residues, banned additives, and other trace substances.

## 1. Introduction

Benzoic acid and benzoate are widely used in food as a kind of preservatives. With the improvement of people’s living standards, more attention has been paid to food safety. The categories and limits of food additives in dairy products are clearly stipulated in National Food Safety Standard: Standard for Uses of Food Additives (GB-2760–2014), in which the limited dosage of benzoic acid is no more than 1.0 g/kg, in which the limited dosage of benzoic acid is no more than 1.0 g/kg [1]. The above food additives can be added to food under the dosage limited by the national standard, but excessive intake of which can be harmful to the human body, especially lead to liver failure, kidney failure, and even cancers [2]. The commonly adopted detection methods of food additives at home and abroad can be divided into two categories: The bioassay method and physicochemical analysis method. Traditional physicochemical analysis method includes High Performance Liquid Chromatography (HPLC), Liquid Chromatograph Mass Spectrometer (LC-MS), and Gas Chromatography, Gas Chromatography-Mass Spectrometry, etc.; And traditional bioassay method includes immunoassay and biosensor. However, the above-mentioned bioassay method and physicochemical analysis method are still limited by many deficiencies in determining various food additives, such as high detection cost, long cycle, low sensitivity, poor reproducibility, complex sample preprocessing, and expensive equipment. Therefore, there is an increasing need to establish a rapid, accurate, and widely applicable method for the detection of ultra-micro food additives [3,4].

THz has been rated as one of the Top Ten Technologies to Change the World by the United States, and has been highly valued by other countries around the world. The high frequency band of the terahertz spectrum overlaps with the far infrared band, and the low frequency band overlaps with the microwave band. Using the terahertz frequency band to detect has a bright development prospect. Besides, the THz ranges between 0.1–10 THz, and the low-frequency vibration absorption frequencies generated by weak interaction between various organic molecules are all located in the terahertz range. When the THz wave passes through the sample, the sample information is carried by the wave, on which the THz spectrum is gained. By studying the THz spectra of samples, physical and chemical information, such as complex permittivity, absorption coefficient, and refractive index of the tested samples, can be obtained. For this reason, an upsurge of study on THz spectroscopy in the field of nondestructive testing has been realized [5]. In recent years, many scholars have applied THz spectroscopy, a new nondestructive testing technology, in many fields, such as industry, food, agricultural products, and pharmaceuticals, made breakthrough progress. THz technology, as an emerging method, can characterize both the intramolecular and intermolecular vibration modes. Hydrogen bond and van der Waals bond [6] when broken, energy emissions are observed in the THz frequency range, and these intermolecular forces play a very important role in molecular and biomolecules detection.

In recent years, some scholars have applied the THz technology to carry out related research to detect agricultural products. Some substances can be detected by well-developed detection methods (such as tablet pressing method, liquid pool method, and attenuated total reflection (ATR) method) because their intermolecular or intramolecular vibrations correspond to the THz frequency band. At present, substances characterized by absorption peaks in the THz range include antibiotics [7], pesticides, amino acids [8], genetically modified crops [9], prohibited additives, food additives [10], foreign substances in food, and adulterated food. However, with the deepening of the research, it is found that conventional THz spectral detection technologies (such as tablet pressing method, liquid pool method, and ATR method) indeed can be used for qualitative analysis of substances, but their detection accuracies fail to meet the requirements of the national standard for food additive limit. Although the above method can obtain the fingerprint spectrum of the sample in the terahertz band, it can only detect a sample to be detected at a milligram level, so it is not suitable for the micro or trace analysis of the sample [11,12].

A metamaterial is a kind of synthetic composite material, which obtains a singular response to electromagnetic waves by periodically arranging sub-wavelength unit structures artificially. Through the design of structural units in metamaterials, extraordinary physical properties, such as negative refraction, abnormal transmission, and reversed Doppler effects, do not exist in natural materials. Among them, metamaterial absorber refers to the device that can effectively absorb the incident wave at a specific frequency, so that the corresponding transmission and reflection are greatly attenuated or even almost disappeared. The metamaterial resonator is characterized by an effective absorbance of the incident wave at a specific frequency and by changing the corresponding transmission and reflection amplitudes to be greatly attenuated or even almost canceled. In recent years, the design and production process of metamaterials has become more and more perfect, more and more frequency resonance can be realized, thus bringing broad attention to metamaterials. The metamaterials resonator is sensitive to the surface dielectric constant, whose change caused by covering different samples is reflected in the position and amplitude of the THz resonance peak frequency, thus enhancing the ability to capture the micro change in the environment. This characteristic enables THz time-domain spectroscopy (THz-TDS) equipment to conduct more sensitive and convenient analysis and quantitative determination of additives.

Taking four kinds of tetracycline as research objects, Chen Min in 2017 [13] explored the feasibility of THz-TDS technology for the qualitative identification of these four kinds of substances by studying their characteristic THz spectra and analyzing their characteristic absorption peaks. Based on Asymmetric Dual-Wire Resonator (ADWR) and THz-TDS technology, Chen established a method system for detecting tetracycline hydrochloride (TCH) and its degradation products aqueous solution, providing a new idea for the rapid determination of tetracycline and its degradation products residues, and other antibiotic residues in liquid food. In 2018, Xu Wendao [14] established a quantitative detection model based on THz metamaterials and nanotechnology. With the assistance of metamaterials, THz-TDS technology was used in combination with nanomaterials (gold nanoparticles and graphene). This model realizes the rapid detection of protein and pesticide, providing new methods for pesticide residue detection and agro-products and biosensing technology. In 2018, Zhang Huo [15] preliminarily studied the possibility of applying the metamaterial resonator in detecting traditional Chinese medicine (TCM), given that the position of resonant peak generated by the metamaterial resonator in its THz transmission spectrum varies with the surface dielectric. By comparing the red shift of samples with different concentrations, Zhang successfully classified several borneols and sensitively obtained the variation of chlorpheniramine maleate with five concentrations ranging from 2‰ to 10‰.

Food preservatives play a vital role in preventing food from spoiling and prolonging the shelf life, but an excessive dosage of food additives can also cause serious hazards to the human body. Based on the theory of electromagnetics, this research designs a terahertz-enhanced metamaterial resonator that can be used to improve the sensitivity of terahertz technology to the detection of benzoic acid additives. The absorption peak of the terahertz metamaterial resonator was designed to be 1.95 THz. Firstly, the original terahertz spectra of the aqueous solution samples of basic benzoic acid were collected. Secondly, smoothing, MSC, and SG + 1^st^D methods were adopted to preprocess the spectra and to study the better spectral pretreatment methods. Then, UVE and CARS will be used to explore the optimal terahertz band selection method. Finally, PLS and LS-SVM models are established, respectively, to realize the enhanced detection of benzoic acid additives. This research is of great significance for promoting the detection and analysis of trace additives in food, whose results can also serve as a reference to the detection of antibiotic residues, banned additives, and other trace substances.

## 2. Materials and Methods

### 2.1. Design and Fabrication of Metamaterial

#### 2.1.1. Numerical Simulation

In this study, the Finite Difference Time Domain (FDTD) Solutions software has been used to design the structural parameters of the metamaterial resonator. FDTD Solutions is a micro-nano photonic simulation software based on finite difference time domain method to solve the vector Maxwell equation. The software can simulate the interaction between electromagnetic waves (from the ultraviolet to terahertz and microwave bands) with typical and complex sub-wavelength structures; it can also be used in the design, analysis, and optimization of micro-nano optical materials, micro-nano photonic devices, etc. Considering that metamaterials are composed of periodic structural units, the transmission or reflection properties of a periodic structural unit in metamaterials can represent the spectral properties of the whole metamaterial. Therefore, in the simulation process, a periodic structural unit of the simulated metamaterial can be used to improve the simulation speed. Full-wave simulation of micro-structure is performed by commercial software FDTD, Lumerical FTDT Solutions 2020. Because of the good conductivity of gold at the terahertz (THz) band, it was regarded as a perfect electrical conductor (PEC) in simulation. The periodic boundary conditions are employed in x and y direction and perfect matched layer (PML) in z direction. A plane wave source is set at 100 μm above the metal structure. To reduce the reflection signal from the back of the quartz substrate, the power monitor is set inside the substrate, 100 μm below the metal structure. The refractive index of quartz substrate is obtained from reference [16].

The absorption peak of benzoic acid at the terahertz is found to be 1.95 THz through the previous study [10]. Therefore, FDTD is used in this paper to design a metamaterial structure that makes the absorption peak close to 1.95 THz. Specific dimensional parameters are shown in Figure 1a: The period (*p*) is 32 μm; the silicon substrate is adopted (the refractive index is 3.335); the coating metal is gold, with cross length (*L*_1_) of 21 μm; length of the cross frame (*L*_2_) is10 μm; width of the cross frame (*W*) is 3 μm. Figure 1b is the spatial distribution of electric field intensity around the “X” shaped array calculated using FDTD, where X and Y are the position coordinates of a unit cell in a metamaterial array. The color from blue to red corresponds to the change of electric field intensity from small to large-that is, the dark red with the largest wavelength represents the largest electric field intensity at this position. The largest electric field is located in the “X” shaped intersection region. This location, also known as a terahertz hotspot, tends to produce a large signal enhancement in terahertz studies. The analysis of the simulation results can provide a theoretical basis for selecting the optimal excitation frequency. Figure 1c is the transmittance spectrum of the metamaterial. The black line is the transmittance spectrum line of the metamaterial calculated using the FDTD method, and the red line is the transmittance spectrum line of the metamaterial by the experimental method, which shows a good agreement with simulation.

#### 2.1.2. Device Fabrication

The 500 μm quartz substrates are cleaned by acetone, methanol, and isopropyl alcohol sonication bath each for 10 min and baked on a hot plate for 2 min. Then, the ultraviolet (UV) photoresist AZ5214 is spin coated on the substrate at 4000 rpm for 40 s and baked at 90 °C for 6 min. Next, the micro-structures were patterned on the AZ5214 by standard UV lithography. After the development process, a 100 nm-thick Au is deposited on the samples by electron beam evaporation tool. Finally, a lift-off process is performed by acetone sonication bath for 20 min to get the metal patterns. Figure 2 is the optical microscopy of metamaterial structure.

### 2.2. Determination of Benzoic Acid Concentration in Solution Based on Metamaterials

#### 2.2.1. Sample Preparation

Benzoic acid was purchased from Aladdin (Shanghai, China) and is delivered in the shape of a white crystal. Deionized water was used as a solvent to reduce the interference of external substances in the experimental results. For the rapid and even dissolution of benzoic acid crystals, the crystals were first fully ground in a mortar, and then benzoic acid aqueous solution samples were prepared with deionized water in turn according to the sample ratio table for preparing an aqueous solution. The concentration ratios are as follows: 0 g/L, 0.055 g/L, 0.11 g/L, 0.215 g/L, 0.635 g/L, 1.115 g/L, 1.985 g/L. Six groups per concentration gradient of the total seven were prepared, and the prepared sample solution was placed on the scroll oscillator for mixing and vibration for 3 min so that benzoic acid can be fully dissolved in the deionized water. Since water has strong absorption of terahertz waves, it is necessary to dry the sample dropped on the metamaterial device. Firstly, the sample solution of 20 μL was absorbed by pipetting gun and dropped on the metamaterial device. Then the device was put into the drying box for 20 min, where the temperature was 50 °C to ensure that the solution on the metamaterial sheet could be fully dried. As the drying was completed, the device was put into the THz system for measurement using the transmission mode, and the THz spectrum of the corresponding sample was collected. All samples were operated in the same way in turn.

#### 2.2.2. THz Spectral Detection System

The detection device adopted in the experiment is the TAS7500 THz time-domain spectrometer produced by Advantest company (Tokyo, Japan). The schematic diagram of the device used in the experiment is shown in Figure 3. The device consists of two ultra-short pulse fiber lasers with a pulse center wavelength of 1550 nm, a maximum output power of 50 mW, and a system scan sampling rate of 8 ms /time. The measuring range of the spectrum is 0.1–5.0 THz; the resolution is 7.6 GHz; scanning frequency is 4048 points per sample; in the range of 1.0–3.0 THz, the light spot diameter is ranged from 2200 μm to 733 μm. During the experiment, the humidity was kept below 10%, and the temperature was 25 °C. In order to reduce the impact of random errors on the experimental results, the measurements were repeated 5 times for each sample, and the average spectrum was obtained for the establishment of subsequent models. The THz spectra of all samples collected were divided into calibration sets and prediction set by K-S (Kennard-Stone) algorithm [17] in a ratio of about 3:1.

#### 2.2.3. Extraction of THz Spectral Parameters

A fast Fourier transform (FFT) was adopted to acquire the spectral distribution of the THz pulse in the frequency, as shown in Equation (1):(1)$$E(\omega )=A(\omega )e^{-i\varphi (\omega )}=\int E(t)e^{-i\omega t}dt$$
(2)$$n(\omega )=\frac{\varphi (\omega )c}{\omega d}+1$$
(3)$$\alpha (\omega )=\frac{1}{d}ln\frac{A_{R}}{A_{S}}$$
where A(ω), φ(ω), E(t) and E(ω) are amplitude, phase of the electric field, time-domain waveform, and frequency-domain waveform, respectively [18].

As shown in Equation (2), ω represents the frequency; φ(ω) is the phase difference between reference signal and sample signal; d is the sample thickness, and c is the speed of light in vacuum. Equation (2) is the calculation of the refractive index of the test sample, while Equation (3) is the formula for sample absorption coefficient, where $A_{S}$ and $A_{R}$ were the amplitude of the sample and reference signals, respectively.

Then the transmission spectrum could be extracted by comparing the sample spectrum with the reference spectrum.
(4)$$T={\left(A_{S}/A_{R}\right)}^{2}$$
where $A_{S}$ and $A_{R}$ were the amplitude of the sample and reference signals, respectively. The averaged transmission spectrum from three measurements for each sample was used for further analysis [19].

The limit of detection (*LOD*) is calculated according to the standard error of predictive concentration, and the slope of the fitting curve between the true value and the predicted value of the model built by terahertz spectrum. The calculation result shows that the *LOD* has a confidence interval of 99.86% [20]. In Equation (4), σ refers to the standard error of predictive concentration; m refers to the slope of the fitting curve, and RMSEP is equal to the value of σ in the model.
(5)$$LOD=\frac{3\sigma }{m}$$

#### 2.2.4. Evaluation of the Benzoic Acid Additive Detection Model

Through comparison of Correlation Coefficient of the Calibration (R_c_), correlation coefficient of the prediction (R_p_), Root Mean Square Error of Calibration (RMSEC), and Root Mean Square Error of Prediction (RMSEP), the models were evaluated. The value of correlation coefficient of the model is inversely proportional to that of the RMSEP, but is in direct proportion to the model’s accuracy. Besides, if values of RMSEC and RMSEP get closer, then the model will be more stable correspondingly.

## 3. Results and Discussion

### 3.1. Spectra Analysis

Terahertz spectrum is rich in information, including many THz optical parameters, such as absorption coefficient, dielectric constant, refractive index, phase angle, etc., which can reflect the internal information of substances in multiple dimensions. Figure 4 shows the THz absorption coefficient spectra of benzoic acid solutions with different concentrations. The spectra were measured by metamaterials. Given that both the front end and the back end have seen more noise interference than other places, the THz spectra with the frequency range of 1.0–3.0 THz were intercepted for the convenience of later data processing. It is shown in Figure 4 that as the concentration of benzoic acid increases, the peak has a tendency to shift to the low frequency and also decreases. The reason may be that the increase of the concentration improves the surface reflection, thereby reducing the absorption peak. The metamaterial resonator has a strong absorption at a specific frequency of the THz waveband (the absorption frequency of the metamaterial resonator used in this paper is 1.95 THz). And the THz metamaterial resonator has a strong interaction with the incident THz wave, which can form a strong electric field. Compared with the direct detection of samples by terahertz waves, the interaction between the electric field which is generated by the metamaterial resonator and the sample is obviously enhanced, which can amplify the sample signal. After the sample was added to the surface of the metamaterial resonator, the surface dielectric environment of the metamaterial resonator was changed, and the impedance matching relationship between the air and the metamaterial resonator structure was also changed, which leads to a red shift in the resonance peak frequency. Therefore, the interaction between the THz wave and the sample can be strengthened through a metamaterial resonator; the frequency shift signal of the resonant peak in the metamaterial resonator can be used for quantitative detection and analyzing the sample.

### 3.2. Validation by FDTD Calculations

Simulation of metamaterial gives us the theoretical analysis of the interaction between THz wave and matter, which is the theoretical guide when we are doing an experiment. The unit cell of metamaterial can be regarded as a micro-resonator comprised of capacitive and inductive response. It will excite a strong resonance, which will be very sensitive to the background change when illuminating the electromagnetic wave at the resonance frequency. The simulation is utilized to propose the scheme of concentration detection and the experiment is for demonstrating and verifying the scheme feasible.

The concentration of the solution can be represented by the thickness of benzoic acid, or the distribution densities of solute depends on the concentration of the solution. If the benzoic acid aqueous solution has a high concentration, its solute after drying can fully cover the metamaterial. On the other hand, the solution with a low concentration will form a discontinuous layer of solute after drying. Based on the two situations mentioned above, two solute film models (1. Various thickness with certain refractive index; 2. Fixed thickness with different refractive index) are made to analyze the influence of concentration on transmission spectrum. The FDTD software was used to conduct these two kinds of simulations. The thickness of the coating was fixed at T = 2 μm, and the variation of the transmission spectrum with the refractive index of the coating is shown in Figure 5a. The refractive index of the coating was fixed at *n* = 1.7, and the variation of transmission spectrum with the thickness of the coating is shown in Figure 5b.

Here, the LC resonance model $f=1/\left(2\pi \sqrt{LC}\right)$ is used to qualitatively analyze the change of metamaterial resonance response with the background change with the above two models. In our paper, each unit of metamaterial can be regarded as a micro-resonator comprised of capacitive and inductive response. The metal strip of the cross can be seen as micro-inductance, which will generate induced current, and the gap between the metal strips forms micro-capacitance, which will generate displacement current under THz wave illuminance. When the cladding of metal micro-structure changes, the cladding’s permeability and permittivity will be changed, which will lead to the inductive and capacitive response change, respectively. Except for some magnetic materials, like ferromagnetic materials, most of the materials have a minor differences in permeability, including air and benzoic acid used in this work. The same permeability of different cladding materials (air and benzoic acid) will lead to a similar inductive response of the micro-structures. However, the difference of permittivity (permittivity is the square of the refractive index, ε = *n*^2^) of distinct materials is usually obvious. Thus, we only analyze the capacitive response in our micro-structures.

In the first situation, as Figure 5a shown, different concentrations of the solution will be transformed into the different distribution densities of solute. The discontinuous layer of benzoic acid can be seen as a uniform continuous film with a refractive index between air and itself. With the refractive index of the background increase, capacitance of micro-structure will be enlarged (C∝ε), and the resonance frequency will be red shift. As Figure 5a shown, the increase of the refractive index of the cladding layer will lead to the resonance peak shift to low frequency.

In the second situation, the various concentration of the solution transforms into the different thickness of solute. Although the metal structure is fully covered by the solute, the thin film cannot cover all evanescent wave fields around the structure, some displacement current between metal strips will still cover the solute film and air. With the film thickness increase, the ratio of film to air in evanescent wave region will be increased, and also the equivalent refractive index around the metal structure which will lead to the capacitance of structure enlarged and resonance peak red shifted, as shown in Figure 5b.

It is worth noting that the increase of the refractive index of the thin film in the first situation will cause the almost linear red shift of the resonance peak, as shown in Figure 5a. In contrast to this situation, the equal step of increment of thickness with the certain refractive index will lead to a minor difference when the film is thick. That is because of the evanescent wave is exponential decay along the z axis (perpendicular to the metamaterial plane), the film closes to the metal structure has more impact on resonance. With fixed thickness, the background of metal structure has the same impact on the evanescent wave, and the equal refractive index difference has an almost equal influence on resonance. However, the impact on capacitance will decrease with the equal step increment of film thickness with fixed refractive index. When increasing the thickness further, the film will cover all evanescent fields around the metal structure, and the capacity response will not be changed, and the frequency of resonance peak will be fixed. The concentration we utilized in our experiment is likely to be the first situation; the equal increment of concentration has a similar impact on resonance peak. The theoretical guide is given by the simulation that the metamaterial composed by the micro-resonator array is more sensitive to concentration change of solution with a low concentration than that with a high concentration.

### 3.3. Preprocessing of THz Spectra of Benzoic Acid Aqueous Solution Based on Metamaterials

In order to eliminate the influence of factors, such as uneven sample mixing, stray light, experimental environment, and noise during sampling, preprocessing methods of SG (Savitzky-Golay) convolution smoothing, multiple scattering correction (MSC), and SG + 1^st^D were selected in this paper to preprocess spectral data [21]. Noises can be reduced by using S-G smoothing, to increase the SNR (Signal to Noise Ratio); 1^st^D can significantly remove baseline interference and other background interferences, resolve overlapping peaks and improve resolution and sensitivity. The preprocessed spectral data and the benzoic acid content in aqueous solution were taken as the input variable and the output variable of the model, respectively, to establish the PLS model. The optimal preprocessing method was then picked out with the reference of the R_p_ and RMSEP in the model. Figure 6 shows the THz spectra of the samples after being preprocessed by SG + 1^st^D.

Results of the PLS model of THz spectra for the samples established by different preprocessing methods are counted in Table 1. The results show that the PLS model established after preprocessing by SG + 1stD has the best results, and the R_p_ and RMSEP of the model are 0.9791 and 1.03×10^−4^, respectively. Therefore, the data preprocessed by SG + 1^st^D are used for further analysis.

### 3.4. THz Wavelength Selection of Benzoic Acid Samples Based on Metamaterial

#### 3.4.1. UVE Algorithm-Based Wavelength Variable Selection

The selection results by UVE of THz spectral variables of benzoic acid samples are shown in Figure 7. The green vertical bar in Figure 7 represents the wavelength separation line, on the left of which is the distribution curve of the wavelength stability. The two horizontal points and lines correspond with two values that represent the threshold values screened by the UVE wavelength method, and the upper and lower threshold values are negative to each other, +9.1851 and −9.1851, respectively. Only those values of stability beyond the range between two threshold values can be used as the input variables of the model. After UVE selection, 141 out of the 340 spectral variables (41.47%) were left, thus reducing the input variables of the model to some extent [22].

#### 3.4.2. CARS Algorithm-Based Wavelength Variable Selection

Competitive adaptive weighted sampling (CARS) refers to an emerging but effective variable selection method. Its principle is identical to that of “the fittest survive” in the evolution theory initiated by Darwin. Each wavelength variable is considered as one individual, while the unsuitable individuals are gradually eliminated. During the selection of wavelength, the wavelength variables with the larger absolute value of regression coefficient in the PLS model were selected by the adaptive reweighted sampling (ARS), and those with smaller weight were removed. Finally, the subset of wavelength variable whose RMSECV value was the lowest was selected by cross-test (CV). This algorithm can be used to effectively single out the best wavelength combination that is relative to detection indexes [23,24].

The CARS algorithm was adopted to select variables on the full THz spectra of flour-benzoic acid mixed samples, and the selection results are shown in Figure 8. From Figure 8a, as the number of sampling increases, the number of retained wavelengths decreases in lockstep. At the outset, the number of retained wavelengths decreases rapidly as the number of sampling increases. Then the reduction gradually slows down, and the number of retained wavelengths ends up remaining unchanged with the increase of the number of sampling. This reflects the process from rough selection to careful selection of wavelength variables by the CARS algorithm. Figure 8b shows the tendency figure of the RMSECV value changing with the increase of the number of sampling in selecting wavelength variables. It can be seen from Figure 8b that RMSECV value gradually decreases when the number of sampling increases from 1 to 28. When the number of sampling reaches 28, RMSECV is at the minimum value of 1.6547 × 10^−4^; when the number of sampling exceeds 28, the RMSECV value gradually increases. The above process indicates that when the number of sampling was below 24, spectral information unrelated to benzoic acid content was weeded out by the CARS algorithm; when the number of sampling was above 24, important information related to benzoic acid content was weeded out. Figure 8c shows the wavelength of the variable regression coefficient changing with the increase of the number of sampling in the wavelength selection process. “*” mark in Figure 8c represents the corresponding number of sampling with the minimum RMSECV value. It can be seen from Figure 8c that when the number of sampling is 28, RMSECV has the minimum value. Correspondingly, the number of wavelength variables is 20.

### 3.5. PLS and LS-SVM Models Establishment and Comparison for THz Spectra of Benzoic Acid Samples in Solution Based on Metamaterials

#### 3.5.1. Establishment of PLS Model for THz Spectra of the Samples

Partial least squares (PLS) perfectly integrates multiple linear regression, principal component analysis, and canonical correlation analysis. When the spectral array and concentration array are decomposed simultaneously, the mutual relationship between them is also considered in the decomposition. With an enhanced corresponding calculation relationship, the optimal correction model can be obtained [25]. Table 2 compares the modeling effect of the PLS model through adopting different methods for selecting wavelength variables. The wavelength variables selected by UVE and CARS were used as input and the benzoic acid content in flour as output to establish the PLS model. To better compare the influence on modeling using various methods to select wavelengths, data of variables of wavelength relatively selected by UVE and CARS were used to establish the PLS models.

#### 3.5.2. Establishment of the LS-SVM Model for THz Spectra of Samples

Least squares support vector machine [26,27] (LS-SVM) refers to a machine learning method that emerged from the basis of statistical learning theory. Its key parameter indexes are input vector, type of kernel function, and the corresponding parameters of this function. The radial basis function (RBF) and linear kernel function (Lin) represent the two most typical kernel functions in LS-SVM. Table 3 compares the performance of the model established by different wavelength variable selection methods and the LS-SVM method. The LS-SVM model was established in which the inputs were wavelength variables selected by the methods of UVE and CARS, and those variables singled out by the methods of UVE and CARS in combination with the principal component analysis method, respectively.

#### 3.5.3. Performance Comparison of PLS Model and LS-SVM Model

Figure 9 shows the optimal predictive value effect of the PLS model and LS-SVM model for benzoic acid concentration of mixed samples. For the PLS model, wavelength selection by CARS not only reduces the calculation amount, but also improves the accuracy of the model. In the establishment of the LS-SVM model, wavelength selection using the CARS method can simplify the model while improving the accuracy of the model. In contrast, the prediction accuracy of the LS-SVM model was higher, and the prediction effect of the CARS-LS-SVM model established through CARS wavelength selection was better, with the prediction correlation coefficient (R_p_) of 0.9953, the prediction root mean square error (RMSEP) of 7.3 × 10^−6^, and the LOD of 2.3610 × 10^−5^ g/mL

## 4. Conclusions

In this paper, the absorption peak of 1.95 THz was designed according to the electromagnetic theory, which is used for the effective detection of benzoic acid samples. The original THz spectra of benzoic acid aqueous solution samples based on metamaterial were preprocessed by SG + 1^st^D, and then the effective spectral wavebands were selected by CARS to establish the model. The results show that the CARS-LS-SVM model established by the method of CARS has an optimal prediction effect, with the prediction correlation coefficient (R_p_) of 0.9953, the prediction root mean square error (RMSEP) of 7.3 × 10^−6^, and the LOD of 2.3610 × 10^−5^ g/ mL. Compared with the traditional compression method, the detection accuracy is improved by about 1400 times [10]. The results provide a basis for terahertz spectral analysis of benzoic acid additives, which will promote the rapid detection of benzoic acid additives in food. This research is of great significance for promoting the detection and analysis of trace additives in food, whose results can also serve as a reference to the detection of antibiotic residues, banned additives, and other trace substances.

## Figures and Tables

**Figure 1 sensors-21-03238-f001:**
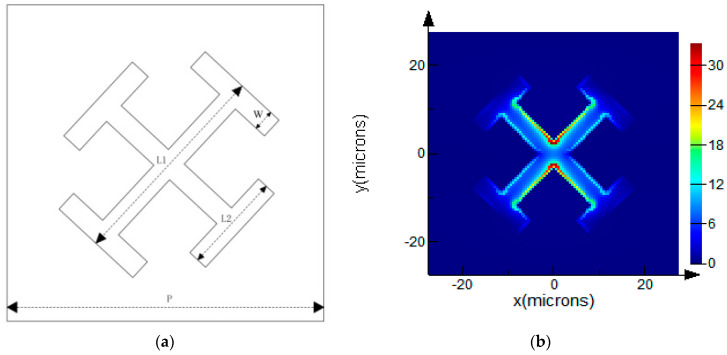
The metamaterial structure and the electromagnetic field distribution of the metamaterial structure based on FDTD method: (**a**) Size structure of the “X” shaped metamaterial; (**b**) electromagnetic field distribution of the “X” shaped metamaterial structure; (**c**) The transmittance spectrum of the “X” shaped metamaterial.

**Figure 2 sensors-21-03238-f002:**
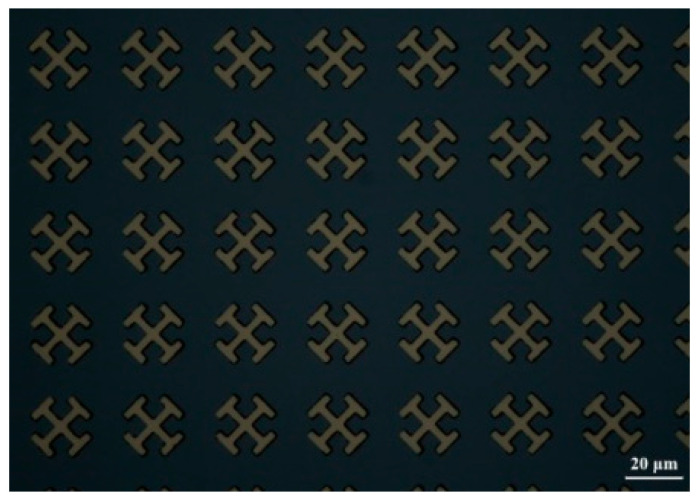
Optical microscopy of the “X” shaped metamaterial structure.

**Figure 3 sensors-21-03238-f003:**
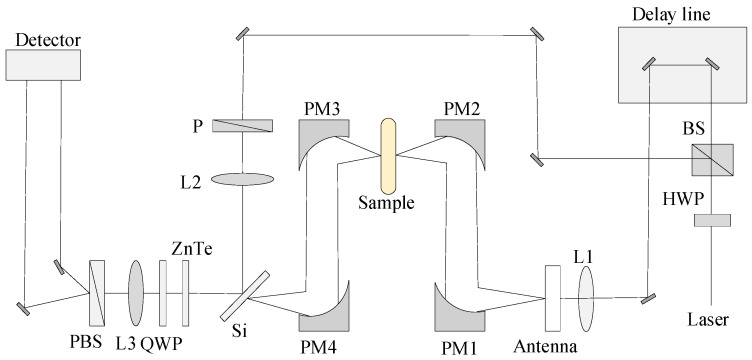
Schematic diagram of THz device.

**Figure 4 sensors-21-03238-f004:**
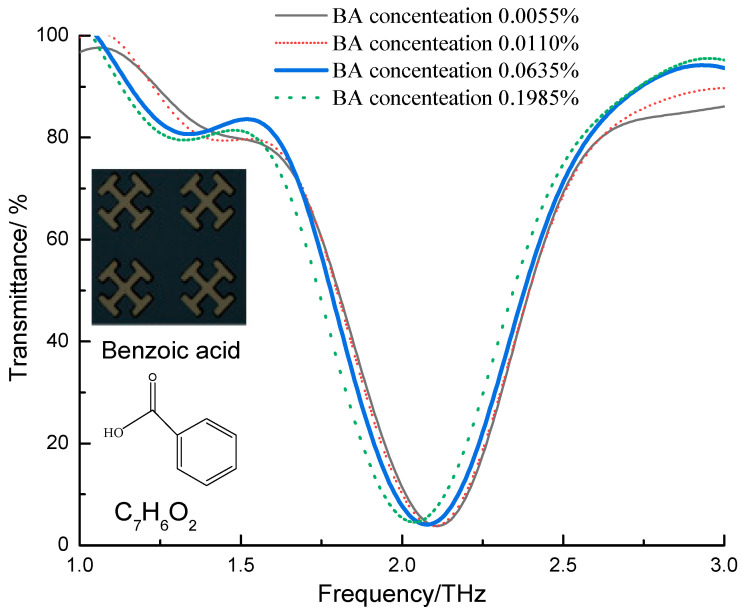
Spectral line of THz absorption coefficient with different benzoic acid concentrations.

**Figure 5 sensors-21-03238-f005:**
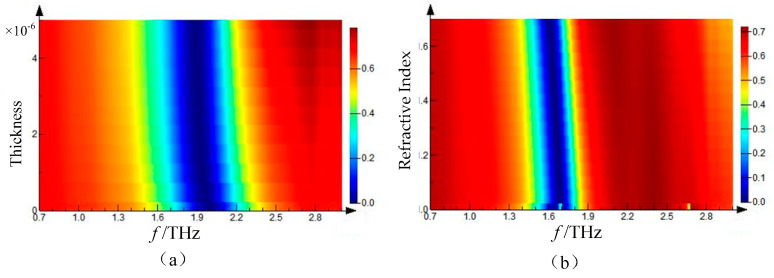
(**a**) Variation of transmission spectrum with the thickness of the coating—the thickness of the coating is fixed at T = 2 μm; (**b**) Variation of transmission spectrum with the refractive index of the coating—the refractive index of the coating is fixed at *n* = 1.7.

**Figure 6 sensors-21-03238-f006:**
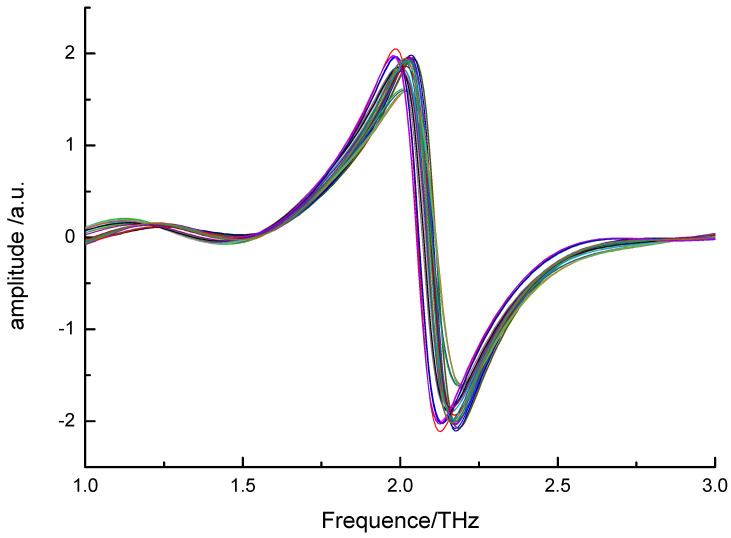
THz Spectra of the samples after preprocessed by SG + 1^st^D.

**Figure 7 sensors-21-03238-f007:**
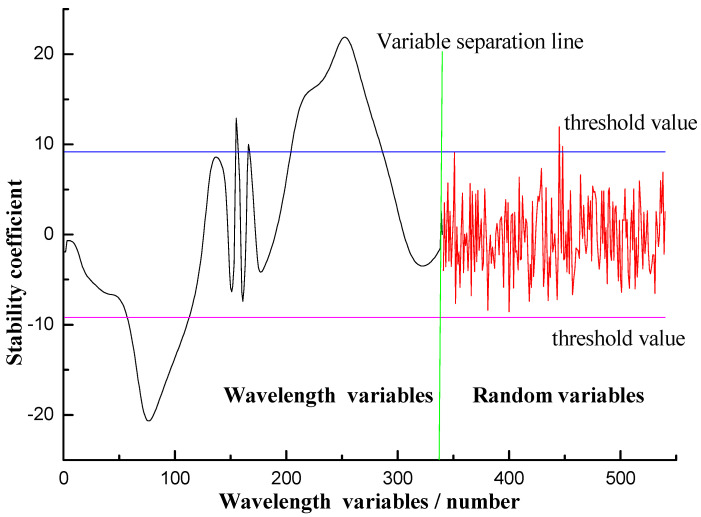
THz spectral wavelength selection by UVE of benzoic acid aqueous solution samples.

**Figure 8 sensors-21-03238-f008:**
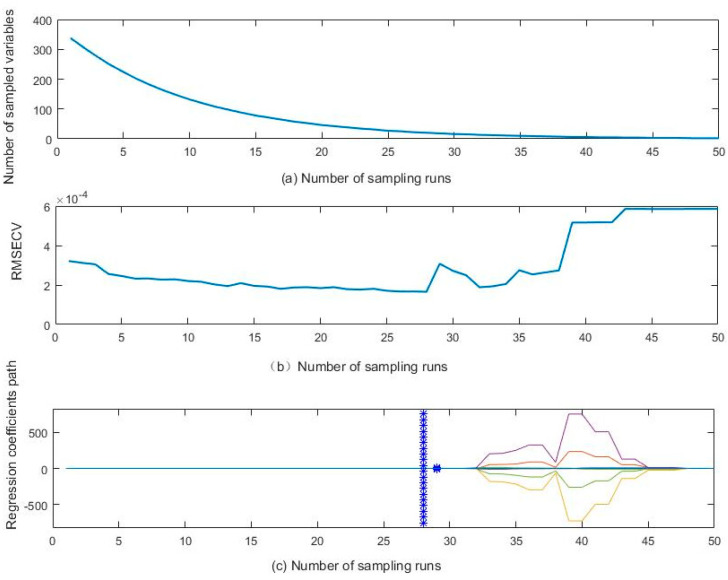
Variable selection results of full THz spectrum of benzoic acid aqueous solution samples using CARS algorithm. (**a**) is the selection results of the mixed sample of wheat flour and BA by CARS algorithm; (**b**) shows the trend plot of RMSECV values corresponding to increasing number of sampling runs; (**c**) shows a trend diagram in which the regression coefficient of the wavelength variable changes with the increasing of sampling runs in the wavelength selection process.

**Figure 9 sensors-21-03238-f009:**
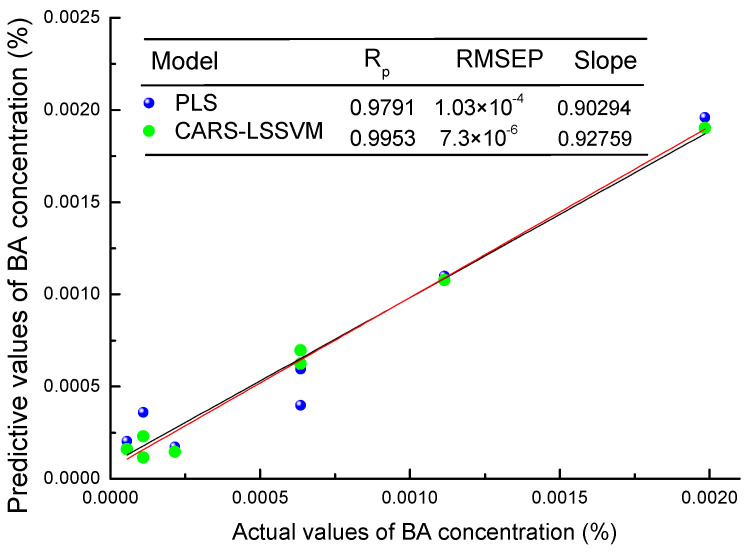
The optimal predictive value effect of PLS and LS-SVM models for benzoic acid concentration in mixed samples.

**Table 1 sensors-21-03238-t001:** Results of the PLS model of THz spectra by different preprocessing methods.

Preprocessing Method	PC	R_c_	RMSEC	R_p_	RMSEP
None	9	0.9744	1.66 × 10^−4^	0.9618	1.21 × 10^−4^
Smoothing	9	0.9729	1.71 × 10^−4^	0.9674	1.17 × 10^−4^
MSC	10	0.9764	1.60 × 10^−4^	0.9765	1.19 × 10^−4^
SG + 1^st^D	10	0.9823	1.39 × 10^−4^	0.9791	1.03 × 10^−4^

**Table 2 sensors-21-03238-t002:** Modeling effects of the PLS models by using various methods for selecting wavelength variables.

Model	Variable Selection Methods	Number of Variable	PC	R_c_	RMSEC	R_p_	RMSEP
PLS	Original data	340	10	0.9823	1.39 × 10^−4^	0.9791	1.03 × 10^−4^
	UVE	141	10	0.9783	1.52 × 10^−4^	0.9197	1.40 × 10^−4^
	CARS	20	9	0.9855	1.19 × 10^−4^	0.9757	1.42 × 10^−4^

**Table 3 sensors-21-03238-t003:** LS-SVM models performance by using various methods for selecting wavelength variables.

Wavelength Selection Methods	No. of Variable	RBF-Kernel			Lin-Kernel		
		γ, σ^2^	R	RMSEP	γ	R	RMSEP
Full spectrum	340	3.1234 × 10^5^, 8.8257 × 10^5^	0.9896	1.15 × 10^−5^	0.2391	0.9802	1.60 × 10^−5^
CARS	20	3.0060 × 10^3^, 50.6610	0.9953	7.3 × 10^−6^	3.3290 × 10^7^	0.9058	2.97 × 10^−5^
UVE	141	1.5350 × 10^5^, 307.0402	0.9925	8.414 × 10^−5^	90.8532	0.9844	1.30 × 10^−5^

## Data Availability

Not applicable.

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
