# Peer review of "Research on Enhanced Detection of Benzoic Acid Additives in Liquid Food Based on Terahertz Metamaterial Devices"

_sensors, 2021, doi:10.3390/s21093238_

Round 1

Reviewer 1 Report

1. The article is interesting but needs revision of English and some concepts need to be explained more clearly.

2. The article has an interesting description of the methods used in the preparation of the samples. Anyone who wants to replicate the results in the article can certainly do so, provided they have  the laboratory tools similar to those used by the authors. 

3. Section 2.2.4 "Evaluation of the Model" doesn't look right. For a start it is very small. But the problem is that it is not  clear at all. It needs clarification regarding what is Rc, Rp, RMSEC and RMSEP. These concepts first appeared in this section and this needs to be substantiated.

4. It is not stated in figure 1b what the colour scale represents. This needs to be corrected. The legend says "Electromagnetic Field Distribution of Metamaterial Structure". This is far from accurate and explanatory of what the figure may represent.

5. I couldn't see the point of showing part b of figure 2. From my point of view it can be removed because it has no interesting information. Alternatively, the authors can better explain why they find it interesting to present this figure.

6. Section 2.2.3 Extraction of THz Spectral Parameters needs revision:

    a) line 223 - This scentence needs correction : "As shown in the Eq.(2), represents the frequency; refers to the extinction coefficient"

    b) The extraction process is not well explained. The authors point to reference 17, but I could not understand how they  
       obtained expressions 1, 2 and 3 from the reference.

    c) They do not say what L is in expressions 2 and 3, but a d appears, possibly representing L.

7. Figure 5 needs color scale and the quantity being plotted could be clearly identified.

8. The analogy the authors make with the LC circuit in line 277 seems to be an oversimplification. According to the FDTD simulations, an increase in refractive index is associated with a decrease in resonance frequency (Fig. 5a). This led the authors to make the said analogy. However, if we think in the same way with Fig. 5b we can see that an increase in thickness seems to indicate also a reduction in resonance frequency. Can the authors explain that with the same model? Please clarify.

9. There are FDTD simulations and experimental results in this work. What is not obvious is what the authors' intent is in showing both. The FDTD simulations do not seem to be used to corroborate the experimental results. They simply appear in the paper as if to fill it out. No comments were made to explain what is the advantage of having these simulations in the paper. I think this could be improved in the paper.

------------------------------
Improvements in figures needed
------------------------------

1. In figure 1a, the colours of the background and the structure are very identical. You need to increase the contrast.

minor corrections
-----------------

line 17 - "1.95 THz" in place of "1.95THz"
line 36 - "1.0 g/kg" in plance of "1.0g/kg"

line 155 - "32 um" instead of "32um"
line 157 - "21 um" in place of "21um"
line 157 - "is 10 um" instead of "is 10um"
line 157 - "3 um" in place of "3um"

line 203 - "humidity" in place of "huminity"

line 267 - "Figure 4" instead of "Figure4"

line 284 - "Figure 5" instead of "Figure5"

# English correction needed #

line 85 - In this line remove the word detection because it becomes repetitive and unnecessary : "as water has strong
absorption in the THz waveband, THz technology is limited in sensing and detecting
detection targets with rich water content [11,12]." 

line 90 - English must be improved: "Through the design of structural units in metamaterials, extraor-
dinary physical properties, such as negative refraction, abnormal transmission and re-
versed Doppler effect, which natural materials do not have can be obtained."

line 92 - English must be improved: "The metamaterial absorber refers to the device that can effectively absorb the incident wave at a
specific frequency and make the corresponding transmission and reflection greatly attenuated or even almost disappeared"

line 95 - English must be improved: "In recent years, as the design and fabrication
equipment of metamaterial has seen gradual improvement, more and more frequency
wavebands can be adapted, thus bringing the broad attention to metamaterial."

line 97 - Confusing sentence. Consider breaking in two: "The
metamaterial absorber is sensitive to the surface dielectric constant whose change caused by covering different samples is reflected in the position and amplitude of the THz resonance peak frequency, which further enhances the ability of capturing the micro change
in the environment."

line 179 - missing an $a$ in this scentence: "Benzoic acid was purchased from Aladdin (www.aladdin-e.com) and is $a$ white crystal"

line 231 - English must be improved: "In Equation (4), refers to the standard error of predictive concen-
tration; m refers to the slope of the fitting curve, and RMSEP is equal to the value of in the
model."

The above list is not exhaustive, you should review the whole paper carefully.

Reviewer 2 Report

Dear authors,

I've found your study to be well-performed and well described.  I have some  minor comments for improvements, please see directly in the attached pdf.

Best regards

Reviewer 3 Report

Due to the fact that extensive editing of English language and style are really required, the clarity and the soundness of the paper may have been diminished at the first lecture of the reviewer.

Some specific observations have been inserted in the original file (attached), with the intention of improving the main identified issues. However, after a serious revision of English, the reviewer may provide further technical suggestions.

Generally, the research work provides significant scientific contribution in the area of detecting minute quantities of a synthetic additive present in a liquid by means of a modern technique, based on metamaterials electromagnetic parameters change at THz frequencies. The approach is a mixture between experiment and computation, evolves gradually and provides at the end a feasible solution of detection. The authors are kindly advised to carefully revise the paper, and to produce a higher quality  written presentation of the valuable work.

Round 2

Reviewer 3 Report

  • line 23:  Please introduce the abbreviations "PLS and LS-SVM"
  • line 26: please use brackets for the abbreviations - LOD, not for the whole words
  • line 245:  relation (3) is not well observed, please edit it clear
  • Fig. 5 - caption: (a) and (b) cases are inverse; please insert the measurement units on both Ox and Oy axes
  • Fig. 6 - if the amplitude on Oy axis is expressed in arbitrary units, please insert this information in the brackets (a. u.)
  • Fig. 7  - please insert the measurement units on the axes
